# Factors associated with self-reported burnout level in allied healthcare professionals in a tertiary hospital in Singapore

Yao Hao Teo[1], Jordan Thet Ke Xu[1], Cowan Ho[1], Jui Min Leong[2], Benjamin Kye Jyn Tan[1], Elisabeth Ker Hsuen Tan[1], Wei-An Goh[1], Elson Neo[1], Jonathan Yu Jing Chua[1], Sean Jun Yi Ng[1], Julia Jie Yi Cheong[1], Jeff Yi-Fu Hwang[2], See Ming Lim[3], Thomas Soo[3], Judy Gek Khim Sng[2], Siyan Yi[2,4,5,6]*

1 Yong Loo Lin School of Medicine, National University of Singapore, Singapore, Singapore, 2 Saw Swee Hock School of Public Health, National University of Singapore and National University Health System, Singapore, Singapore, 3 Occupational Health Clinic, Department of Medicine, National University Hospital, National University Health System, Singapore, Singapore, 4 KHANA Center for Population Health Research, Phnom Penh, Cambodia, 5 Center for Global Health Research, Touro University California, Vallejo, CA, United States of America, 6 School of Public Health, National Institute of Public Health, Phnom Penh, Cambodia

* ephsyi@nus.edu.sg

**Data Availability Statement:** All relevant data are within the manuscript and its Supporting Information files.

## Abstract

### Background

Burnout has adverse implications in healthcare settings, compromising patient care. Allied health professionals (AHPs) are defined as individuals who work collaboratively to deliver routine and essential healthcare services, excluding physicians and nurses. There is a lack of studies on burnout among AHPs in Singapore. This study explored factors associated with a self-reported burnout level and barriers to seeking psychological help among AHPs in Singapore.

### Methods

We conducted a cross-sectional study in a sample of AHPs in a tertiary hospital from October to December 2019. We emailed a four-component survey to 1127 eligible participants. The survey comprised four components: (1) sociodemographic characteristics, (2) Maslach Burnout Inventory (MBI-HSS), (3) Areas of Worklife Survey, and (4) Perceived Barriers to Psychological Treatment (PBPT). We performed a multiple logistic regression analysis to identify factors associated with burnout. Adjusted odds ratios (AORs) and associated 95% confidence intervals (CIs) were computed.

### Results

In total, 328 participants completed the questionnaire. The self-reported burnout level (emotional exhaustion>27 and/or depersonalization>10) was 67.4%. The majority of the respondents were female (83.9%), Singaporean (73.5%), aged 40 years and below (84.2%), and Chinese ethnicity (79.9%). In the multiple logistic regression model, high burnout level was negatively associated with being in the age groups of 31 to 40 (AOR 0.39, 95% CI 0.16–

**Funding:** The authors received no specific funding for this work.

**Competing interests:** The authors have declared that no competing interests exist.

0.93) and 40 years and older (AOR 0.30, 95% CI 0.10–0.87) and a low self-reported work-load (AOR 0.35, 95% CI 0.23–0.52). High burnout level was positively associated with a work experience of three to five years (AOR 5.27, 95% CI 1.44–20.93) and more than five years (AOR 4.24; 95% CI 1.16–16.79. One hundred and ninety participants completed the PBPT component. The most frequently cited barriers to seeking psychological help by participants with burnout (*n* = 130) were 'negative evaluation of therapy' and 'time constraints.'

## Conclusions

This study shows a high self-reported burnout level and identifies its associated factors among AHPs in a tertiary hospital. The findings revealed the urgency of addressing burnout in AHPs and the need for effective interventions to reduce burnout. Concurrently, proper consideration of the barriers to seeking help is warranted to improve AHPs' mental well-being.

## Introduction

Burnout is a prolonged response to chronic emotional and interpersonal stressors on the job, comprising three dimensions: exhaustion, cynicism, and inefficiency [1]. These dimensions are further defined as follows: exhaustion of emotional or physical capacity due to stress, a degree of indifference or detachment from various aspects of work, and a sense of inadequacy or reduced personal accomplishment, respectively [1].

In healthcare settings, burnout negatively impacts outcomes at the individual, interpersonal, and institutional levels. At the individual level, burnout is associated with reduced job satisfaction, increased absenteeism, medical errors, sickness, injury, and accidents among healthcare providers [2, 3]. These individual-level impacts may lead to reduced care quality and higher mortality among patients [4, 5]. From an interpersonal perspective, burnout is associated with emotional dissonance due to chronic exhaustion and cynicism [6]. Emotional dissonance is described as a conflict between personal emotions and organizational demands. On an institutional level, burnout is linked to a higher turnover of healthcare workers [7, 8] and decreased workforce efficiency [9], posing a substantial economic burden on the healthcare system [10].

The pernicious nature of burnout in healthcare settings has prompted numerous studies on its prevalence in physicians and nurses in Singapore and internationally. For example, high burnout levels and their associated factors among physicians and nurses have been reported in Singapore [11, 12]. Extensive research involves the barriers to seeking help for doctors, such as fear of stigma, lack of available time, and lack of convenient access [13, 14].

Allied health professionals (AHPs) are defined as individuals who work collaboratively to deliver routine and essential healthcare services, excluding physicians and nurses [15, 16]. AHPs include, but are not limited to, occupational therapists, physiotherapists, pharmacists, medical social workers, and radiographers [17]. This system is similarly adopted in the United Kingdom [18] and the United States [19] and plays an essential role in improving hospital efficiencies and access to care [19]. In Singapore, the Allied Health Professions Council (AHPC) defines and classifies allied health occupations similar to other countries [20].

Studies in other countries have reported a high prevalence of burnout in AHPs. In the United States, physiotherapists and occupational therapists reported high rates of emotional exhaustion (58%), negative feelings about their work and their clients (94%), and an almost

non-existent sense of personal accomplishment (1%) [21]. However, there are currently no studies examining burnout levels and their associated risk factors among AHPs in Singapore.

This study aims to identify the self-reported burnout levels and explore their associations with sociodemographic factors and the work environment among AHPs in Singapore. Based on the evidence from studies on doctors and nurses [11, 12, 22], we hypothesized that burnout levels among AHPs in Singapore would be similarly high, and age and work experience would be significantly associated with burnout levels. Our secondary objective is to identify significant barriers in seeking psychological help among AHPs with a high burnout level.

## Materials and methods

### Study design and sampling

We conducted a cross-sectional study among AHPs working in a tertiary acute care hospital between October 2019 to December 2019. Based on previous studies looking at the prevalence of burnout in AHPs and the total number of AHPs in Singapore [23, 24], we determined the sample size through the application of a single proportion formula with the assumption of 60% prevalence, 5% marginal error, and 95% confidence level (CI). The minimum required sample size for the study was 348.

### Inclusion and exclusion criteria

We defined AHPs according to the definition recommended by Singapore's AHPC–all healthcare professionals who work collaboratively to deliver routine and essential healthcare services, excluding physicians and nurses [15, 16]. AHPs in a tertiary hospital of all seniority levels were included in this study [16].

### Questionnaire design and measurement

We developed an electronic survey and emailed all AHP staff working for the tertiary hospital to request their participation. The survey comprised four components: (1) sociodemographic characteristics, (2) Maslach Burnout Inventory (MBI-HSS), (3) Areas of Worklife Survey (AWS), and (4) Perceived Barriers to Psychological Treatment (PBPT).

Sociodemographic questions were adapted from the Singapore National Health Survey 2010 [25], covering residency status, age, gender, ethnicity, income levels, caregiver status, occupation, employment history, physical activity levels, and mental health.

We assessed burnout by using the Maslach Burnout Inventory (MBI), in particular, the MBI-Human Services Survey for Medical Personnel MBI-HSS(MP) [26]. MBI has been widely used in different settings [27] and is the best-known questionnaire used in most clinical studies assessing burnout [28]. The questionnaire consisted of nine questions under emotional exhaustion (EE), five questions under depersonalization (DP), and eight questions under personal accomplishment (PA). Participants were asked to rate on a Likert scale of 0 (never) to 6 (every day) on how often they experienced the symptoms, and the total scores for each subsection were tallied. Higher EE and DP scores correspond to a higher burnout level, while, conversely, lower PA scores signify a higher burnout level. The scale's validity has previously been demonstrated in similar studies in Japan and China, countries with strong Asian cultural influence [29–31]. It has also been used to evaluate burnout levels in studies in Singapore [11, 32].

The maximum score was 54 points for EE, 30 points for DP, and 48 points for PA. No universal cut-off score has been recommended to define burnout. In a systematic review of burnout among healthcare professionals, burnout was defined using the cut-offs of EE>27 or DP>10, with PA excluded in the majority of the included studies [27]. PA was also excluded

from previous studies because its association with burnout has been more variable and complex [1]. It has been postulated that PA may be a function of EE and DP because a work situation with overwhelming demands may also erode one's PA [1]. Hence, we defined a high burnout level as experiences of a high level of EE (EE>27), DP (DP>10), or both [33]. We also included an analysis of a high burnout level defined according to EE>27, DP>10, or PA<33 (Appendix 1).

The AWS is a 28-item scale that is part of the MBI toolkit [34]. The scale examines the dimensions of an individual's work life and predicts their relationship with burnout [35]. The six dimensions assessed in the survey were: workload, control, reward, community, fairness, and values. "Workload" (five items) refers to the employee's ability to cope with work demands. "Control" (four items) refers to the level of active involvement of an employee in work decisions. "Reward" (four items) refers to rewards that place higher value and recognition on an employee's work. "Community" (five items) refers to the overall quality of social interaction at work. "Fairness" (six items) refers to the general equity of decisions made at the workplace. Furthermore, "Values" (four items) refers to the dissonance between personal and organizational values [36]. Respondents were asked to rate on a Likert scale of 1 (strongly disagree) to 5 (strongly agree) on their perceptions of work setting qualities that play a role in burnout. The item scores in each domain are then averaged. A higher AWS score indicates a more balanced relationship, rather than a conflicted one [35], between the respondent and their work [37].

The last component of the survey comprised the 27-item PBPT questionnaire [38]. Items are classified into nine domains: stigma, lack of motivation, emotional concerns, negative evaluations of therapy, misfit of therapy to needs, time constraints, participation restriction, availability of services, and cost [38]. We asked participants to rate on a 5-point Likert scale the degree to which each item hindered them from seeing a counselor or a therapist. A score of four to five was deemed as "substantial barriers." A domain is deemed to represent a "substantial barrier" if at least one item within that domain was reflected as a "substantial barrier." Given the lengthy questionnaire and to improve the overall response rate [39], we made the PBPT questionnaire component optional for participants in this study.

## Data analyses

We used R Commander version 2.7.11 to perform all statistical analyses. We computed Cronbach's alpha for each MBI subscale and AWS domain to assess reliability. We performed bivariate analyses of the demographic factors and the AWS dimensions to examine their association with burnout level using Pearson's Chi-square test or Fisher's exact tests (when a cell count was smaller than five). We identified factors associated with burnout levels by using multiple logistic regression analysis. We entered variables with statistical significance ($p<0.05$) in bivariate analyses simultaneously in the multiple logistic regression model. For respondents who completed the optional component on PBPT, we recorded the incidence of expressing a variable as a "substantial barrier" among participants who experienced a high burnout level.

## Ethical considerations

The National Healthcare Group Domain Specific Review Board approved this study (2019/ 00477). No identifiable information of participants was collected. We stored all data on RED-Cap, a secure, Health Insurance Portability, and Accountability Act compliant, web-based server. We included a participant information sheet in the email, providing all relevant information on participant anonymity and consent for voluntary participation.

## Results

### Sociodemographic characteristics

Among the 1127 eligible AHPs invited, 345 participated in the survey. However, we excluded 17 questionnaires due to incomplete entries. We included a total of 328 respondents in the analyses, providing a response rate of 29.1%. Compared to those who did not participate, our participants were more likely to be female, non-Singaporeans/non-SPR, 21 to 30 years old, and had more than three years of working experience.

Table 1 shows the sociodemographic characteristics of the respondents. The majority of the respondents were Singaporean (73.5%), aged 40 years and below (84.2%), female (83.9%), and Chinese ethnicity (79.9%). Almost all respondents were working full time (94.2%). More than half of the respondents had worked for more than five years in the same organization. Approximately half of the respondents worked as frontline staff and reported low levels of physical activity. Only a small proportion of the respondents reported a history of mental illness or had sought help from a professional within the past year for mental illness. The Cronbach's alpha coefficients for EE, DP, and PA in MBI-HSS in this study were 0.93, 0.81, and 0.85, respectively, suggesting that the overall measurement was reliable.

### Burnout level and associated sociodemographic factors

The self-reported burnout level among AHPs in this study was 67.4%. A majority of the respondents reported a high burnout level on EE ($n = 203$, 61.9%), less than half reported a high level on DP ($n = 139$, 42.4%), and more than one-third had both high EE and DP ($n = 122$, 37.1%). Among the occupational groups, dieticians (94.7%) and pharmacists (82.5%) had the highest burnout levels.

Table 2 shows the sociodemographic characteristics of AHPs stratified by burnout levels. Full-time workers were significantly more likely to experience a high burnout level than part-time workers. Respondents with more than one year of work experience were significantly more likely to experience a high burnout level than those with less than one year of work experience. Respondents who had sought professional mental help in the past year were significantly more likely to have a high burnout level than those who did not.

### AWS domains and association with burnout levels

The Cronbach's alpha coefficients for workload, control, reward, community, fairness, and values were 0.78, 0.77, 0.89, 0.86, 0.82, and 0.78, respectively. As shown in Fig 1, all AWS domains were significantly associated with a higher burnout level ($p \leq 0.01$), with workload, control, and reward showing the most significant differences in the mean scores between participants with a low and high burnout level.

### AWS individual statements and association with burnout levels

Fig 2 presents the absolute mean score differences of responses to individual AWS statements between participants with a high and low burnout level. The majority of the mean score differences in all domains were significant. The workload domain had the highest absolute difference compared to the other domains. In particular, the statements "I have so much work to do on the job that it takes me away from my personal interests" (question 3) and "I do not have time to do the work that must be done" (question 1) in the workload domain scored the highest absolute difference in mean scores among all questions.

**Table 1. Sociodemographic characteristics of allied health professionals in a tertiary hospital in Singapore.**

| Variables | | Number (*n* = 328) |
|---|---|---|
| | | *n* (%) |
| Residency status | | |
| | Singaporean | 241 (73.5) |
| | Permanent resident | 59 (18.0) |
| | Foreigner | 28 (8.5) |
| Age group | | |
| | 21 to 30 | 137 (41.8) |
| | 31 to 40 | 139 (42.4) |
| | 41 years and above | 52 (15.9) |
| Sex | | |
| | Male | 53 (16.1) |
| | Female | 275 (83.9) |
| Ethnic group | | |
| | Chinese | 262 (79.9) |
| | Non-Chinese | 66 (20.1) |
| Average monthly household income | | |
| | Less than S$5000 | 25 (7.6) |
| | S$5000 to S$9000 | 118 (36.0) |
| | S$9000 and above | 109 (33.2) |
| | Not disclosed | 76 (23.2) |
| Caregiver status | | |
| | Yes | 72 (22.0) |
| | No | 229 (69.8) |
| | Do not wish to disclose | 27 (8.2) |
| Occupation | | |
| | Clinical psychologist | 5 (1.5) |
| | Radiographer | 32 (9.8) |
| | Dietician | 19 (5.8) |
| | Medical technologist | 96 (29.3) |
| | Medical social worker | 19 (5.8) |
| | Occupational therapist | 33 (10.0) |
| | Pharmacist | 40 (12.2) |
| | Physiotherapist | 28 (8.5) |
| | Podiatrist | 5 (1.5) |
| | Respiratory therapist | 5 (1.5) |
| | Speech therapist | 17 (5.2) |
| | Others | 29 (8.8) |
| Duration working at the current organization | | |
| | <1 year | 24 (7.3) |
| | 1–2 years | 47 (14.3) |
| | 3–5 years | 64 (19.5) |
| | >5 years | 193 (58.8) |
| Nature of work | | |
| | Front line staff | 175 (53.4) |
| | Administrator | 11 (3.3) |
| | Junior management | 55 (16.8) |
| | Senior management | 20 (6.1) |

(*Continued*)

**Table 1.** (Continued)

| Variables | | Number (*n* = 328) |
|---|---|---|
| | | *n* (%) |
| | Others | 67 (20.4) |
| Employment status | | |
| | Full time | 309 (94.2) |
| | Part-time | 19 (5.8) |
| Average number of night shifts per month | | |
| | 1–3 times | 33 (10.0) |
| | 4–6 times | 18 (5.5) |
| | 7 or more times | 4 (1.2) |
| | Not applicable | 273 (83.2) |
| Level of physical activity† | | |
| | Low | 185 (56.6) |
| | Moderate | 74 (22.6) |
| | High | 68 (20.8) |
| Previous history of mental illness* | | |
| | Yes | 7 (2.2) |
| | No | 311 (97.8) |
| Sough medical help in the past year | | |
| | Yes | 15 (4.7) |
| | No | 305 (95.3) |

*Mental illness refers to a behavioral or psychological syndrome or pattern in an individual that causes clinically significant distress. It warrants diagnosis and management by a medical professional [40, 41].

† Low physical activity refers to sedentary, little, or no exercise. Moderate physical activity refers to a low level of exertion or aerobic exercises for 20–60 min per week. High physical activity refers to aerobic exercises for > 1 h per week.

## Factors associated with burnout levels

In the multiple logistic regression model (Table 3), AHPs who had lower mean scores in the workload subdomain of the AWS, indicative of a high workload burden, were almost three times more likely to have a high burnout level than those who had higher mean scores. Compared to respondents aged 30 years and below, older AHPs aged 31 and above were significantly less likely to have a high burnout level. Moreover, respondents who had worked in the current organization for more than three years were approximately five times more likely to experience a higher burnout level than respondents who had worked in the current organization for less than one year.

## Perceived barriers to seeking psychological help

Of the total, 57.9% (*n* = 190) of participants completed the optional component on PBPT, of which 130 had a high burnout level. Table 4 shows that, among the participants with a high burnout level, the most frequently cited barriers to seeking psychological help were 'negative evaluation of therapy' (60%) and 'time constraints' (50%).

## Discussion

This study is the first to investigate the self-reported burnout level and its related factors among AHPs in Singapore. We found a high burnout level at 67.4% among AHPs in a tertiary

**Table 2. Sociodemographic characteristics of allied health professionals stratified by burnout levels.**

| Variables | | Low burnout (n = 221) | High burnout (n = 107) | |
|---|---|---|---|---|
| | | n (%) | n (%) | p-value |
| Residency status | | | | 0.10 |
| | Singaporean | 164 (68.0) | 77 (32.0) | |
| | Permanent Resident | 43 (72.9) | 16 (27.1) | |
| | Foreigner | 14 (50.0) | 14 (50.0) | |
| Age group | | | | <0.01 |
| | 21 to 30 | 101 (73.7) | 36 (26.3) | |
| | 31 to 40 | 94 (67.6) | 45 (32.4) | |
| | 41 years and above | 26 (50.0) | 26 (50.0) | |
| Gender | | | | 1.00 |
| | Male | 36 (67.9) | 17 (32.1) | |
| | Female | 185 (67.3) | 90 (32.7) | |
| Ethnic group | | | | 0.14 |
| | Chinese | 182 (69.5) | 80 (30.5) | |
| | Non-Chinese | 39 (59.1) | 27 (40.9) | |
| Average monthly household income | | | | 0.14 |
| | Less than S$5000 | 18 (72.0) | 7 (28.0) | |
| | S$5000 to S$9000 | 85 (72.0) | 33 (28.0) | |
| | S$9000 and above | 64 (58.7) | 45 (41.3) | |
| | Not disclosed | 54 (71.1) | 22 (28.9) | |
| Caregiver status | | | | 0.96 |
| | Yes | 48 (66.7) | 24 (33.3) | |
| | No | 154 (67.2) | 75 (32.8) | |
| | Do not wish to disclose | 19 (70.4) | 8 (29.6) | |
| Occupation | | | | 0.07 |
| | Clinical psychologist | 2 (40.0) | 3 (60.0) | |
| | Radiographer | 19 (59.4) | 13 (40.6) | |
| | Dietician | 18 (94.7) | 1 (5.3) | |
| | Medical technologist | 64 (66.7) | 32 (33.3) | |
| | Medical social worker | 15 (78.9) | 4 (21.1) | |
| | Occupational therapist | 21 (63.6) | 12 (36.4) | |
| | Pharmacist | 33 (82.5) | 7 (17.5) | |
| | Physiotherapist | 17 (60.7) | 11 (39.3) | |
| | Podiatrist | 3 (60.0) | 2 (40.0) | |
| | Respiratory therapist | 2 (40.0) | 3 (60.0) | |
| | Speech therapist | 10 (58.8) | 7 (41.2) | |
| | Others | 17 (58.6) | 12 (41.4) | |
| Duration working at the current organization | | | | <0.01 |
| | <1 year | 9 (37.5) | 15 (62.5) | |
| | 1–2 years | 31 (66.0) | 16 (34.0) | |
| | 3–5 years | 53 (82.8) | 11 (17.2) | |
| | >5 years | 128 (66.3) | 65 (33.7) | |
| Nature of work | | | | 0.98 |
| | Front line staff | 116 (66.3) | 59 (33.7) | |
| | Administrator | 7 (63.6) | 4 (36.4) | |
| | Junior management | 38 (69.1) | 17 (30.9) | |
| | Senior management | 14 (70.0) | 6 (30.0) | |

(*Continued*)

**Table 2.** (Continued)

| Variables | | Low burnout (*n* = 221) | High burnout (*n* = 107) | |
|---|---|---|---|---|
| | | *n* (%) | *n* (%) | *p*-value |
| | Others | 46 (68.7) | 21 (31.3) | |
| Employment status | | | | <0.01 |
| | Full time | 214 (69.3) | 95 (30.7) | |
| | Part time | 7 (36.8) | 12 (63.2) | |
| Average number of night shifts per month | | | | 0.24 |
| | 1–3 times | 25 (75.8) | 8 (24.2) | |
| | 4–6 times | 12 (66.7) | 6 (33.3) | |
| | 7 or more times | 1 (25.0) | 3 (75.0) | |
| | Not applicable | 183 (67.0) | 90 (33.0) | |
| Level of physical activity† | | | | 0.58 |
| | Low | 129 (69.7) | 56 (30.3) | |
| | Moderate | 49 (66.2) | 25 (33.8) | |
| | High | 43 (63.2) | 25 (36.8) | |
| Previous history of mental illness* | | | | 0.43 |
| | Yes | 6 (85.7) | 1 (14.3) | |
| | No | 206 (66.2) | 105 (33.8) | |
| Sough medical help in the past year | | | | <0.01 |
| | Yes | 15 (100.0) | 0 (0.0) | |
| | No | 198 (64.9) | 107 (35.1) | |

*Mental illness refers to a behavioral or psychological syndrome or pattern in an individual that causes clinically significant distress. It warrants diagnosis and management by a medical professional [40, 41].

† Low physical activity refers to sedentary, little, or no exercise. Moderate physical activity refers to a low level of exertion or aerobic exercises for 20–60 min per week. High physical activity refers to aerobic exercises for >1 h per week.

hospital. Based on the job demands-resources model of burnout, high EE and DP scores in our study demonstrates a high probability of resource conservation by AHPs. AHPs may spend less time with patients, resulting in increased clinical errors [42] and negatively impacting

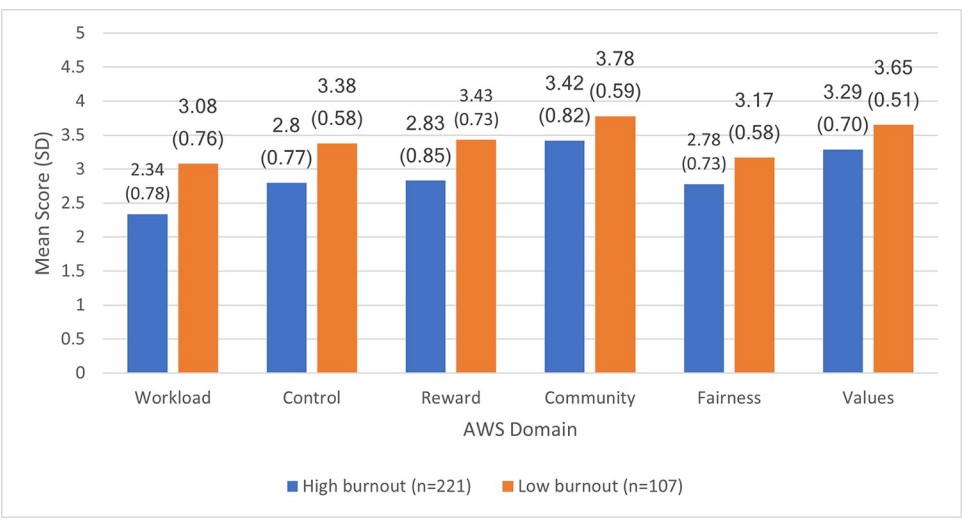

**Fig 1. Comparisons of the mean scores of the Areas of Worklife Survey domains stratified by burnout levels (*n* = 328).**

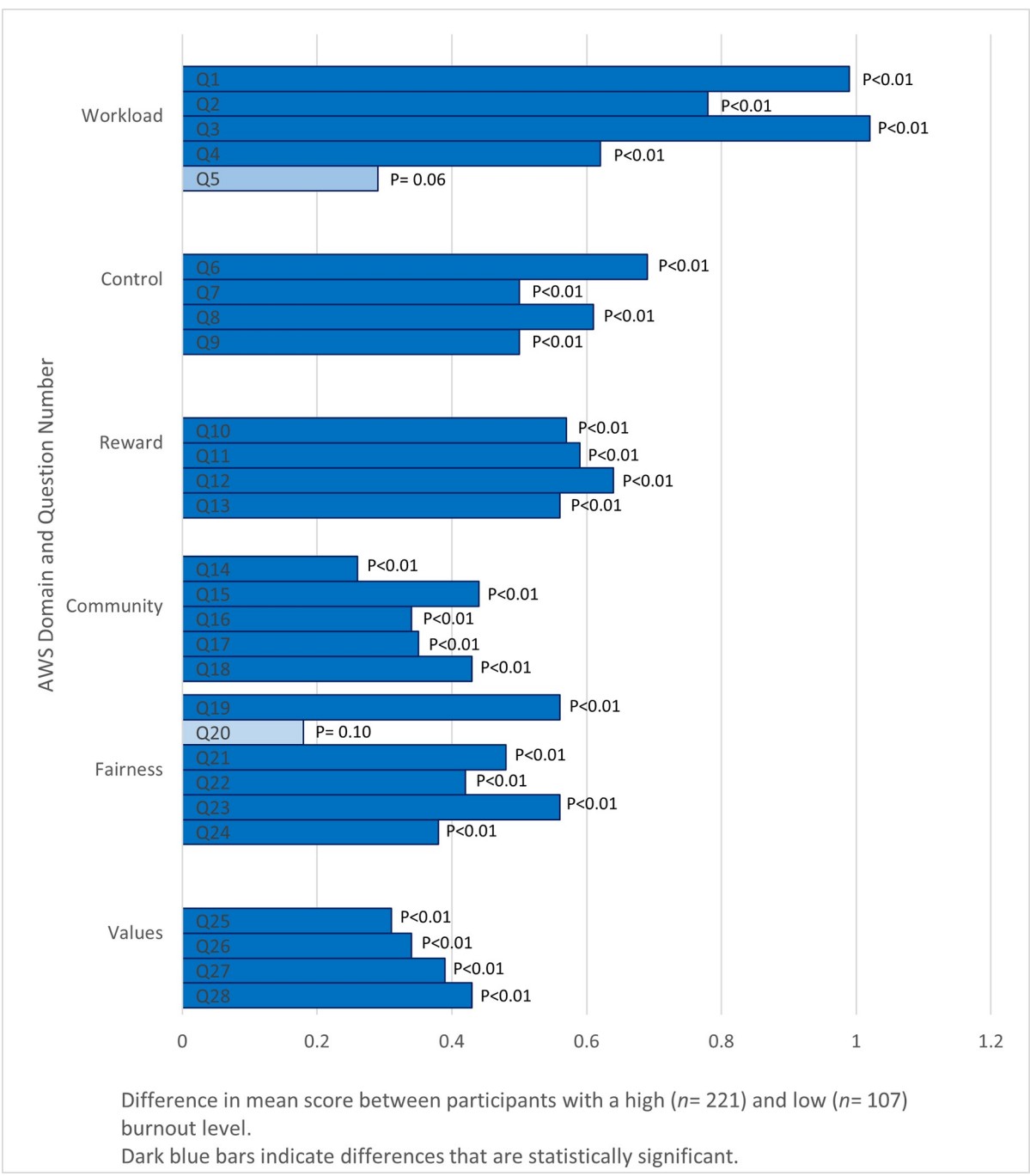

Difference in mean score between participants with a high (*n*= 221) and low (*n*= 107) burnout level.
Dark blue bars indicate differences that are statistically significant.

**Fig 2. Comparisons of difference in mean scores of Areas of Worklife Survey statements in all domains between participants with a high (*n* = 221) and low (*n* = 107) burnout level.**

patient care. However, compared to a study conducted among physical and occupational therapists in the United States, while the EE scores were similar (58% vs. 62% in our study), the DP scores in our study were significantly lower (94% vs. 42% in our study) [21]. The relatively lower depersonalization scores may be attributed to the participants' organizational factors, such as different healthcare systems and attitudes towards work between AHPs in Asian and

**Table 3. Factors associated with burnout levels in a multiple logistic regression analysis.**

| | | Coefficient (SE) | AOR (95% CI) | *p*-value |
|---|---|---|---|---|
| AWS domain | | | | |
| | Workload | -1.05 (0.21) | 0.35 (0.23, 0.52) | <0.01 |
| | Control | -0.46 (0.29) | 0.63 (0.35, 1.10) | 0.11 |
| | Reward | -0.34 (0.25) | 0.71 (0.43, 1.17) | 0.18 |
| | Community | -0.39 (0.26) | 0.68 (0.40, 1.12) | 0.13 |
| | Fairness | -0.18 (0.32) | 0.83 (0.45, 1.54) | 0.56 |
| | Values | -0.10 (0.30) | 0.90 (0.50, 1.63) | 0.74 |
| Age group | | | | |
| | 21 to 30 | Reference | 1.00 | - |
| | 31 to 40 | -0.93 (0.45) | 0.39 (0.16, 0.94) | 0.04 |
| | 41 years and above | -1.20 (0.55) | 0.30 (0.10, 0.87 | 0.03 |
| Duration working at the current organization | | | | |
| | <1 year | Reference | 1.00 | - |
| | 1–2 years | 0.61 (0.68) | 1.84 (0.50, 7.23) | 0.37 |
| | 3–5 years | 1.66 (0.68) | 5.27 (1.44, 20.91) | 0.01 |
| | >5 years | 1.44 (0.68) | 4.23 (1.16, 16.76) | 0.03 |
| Employment status | | | | |
| | Part-time | Reference | 1.00 | - |
| | Full-time | 0.62 (0.60) | 1.86 (0.58–6.42) | 0.30 |
| Sought medical help in the past year | | | | |
| | No | Reference | 1.00 | - |
| | Yes | 17.13 (854.15) | 27398446.44 (NA) | 0.98 |

Abbreviations: AWS, Areas of Worklife Survey; SE, standard error; CI, confidence interval; AOR, adjusted odds ratio.

Western societies [43]. The lower level could be culturally equivalent to the United State's higher levels due to differences in the participants' attitudes towards surveys and response patterns [44–46].

Of note, the high self-reported burnout level in pharmacists (82.5%) and dieticians (94.7%) is concerning. We postulate that pharmacists may be prone to experiencing burnout and lower job satisfaction than other occupations, with more job variety reported in previous studies [47]. However, similar studies have shown that dieticians score lower EE than comparison groups of doctors, nurses, and social workers [23], indicating lower burnout. Hence, the high burnout level among dieticians may be due to other organizational or demographic factors. As

**Table 4. Perceived barriers to seeking psychological help among participants with a high burnout level (*n* = 130).**

| | *n* (%) |
|---|---|
| Stigma | 29 (48.3) |
| Lack of motivation | 16 (26.7) |
| Emotional concerns | 16 (26.7) |
| Negative evaluation of therapy | 36 (60.0) |
| Misfit of therapy to needs | 27 (45.0) |
| Time constraints | 30 (50.0) |
| Participation restriction | 27 (45.0) |
| Availability of services | 25 (41.7) |
| Cost | 21 (35.0) |

the sample size of pharmacists and dieticians in this study was small, these associations were not statistically significant. Further studies will be warranted to identify the associated factors of burnout.

In the multiple regression analysis, we found a higher burnout level in the younger group of 21 to 30 than in AHPs aged 31 years and above. Previous studies have supported this trend of burnout affecting younger employees [2, 3]. The lower burnout level in older participants may be explained by their better coping or occupational handling stress [48, 49]. Work experience may play an essential role in burnout. Employees who have worked for a longer duration (three years and above) in the same organization were more likely to have a high burnout level than those working for less than a year. We postulate that this could be due to long-term exposure to the patient suffering at the workplace, resulting in emotional exhaustion [50, 51].

We found that heavier self-reported workloads are associated with a higher burnout level among the AHPs. It was the only subdomain of the AWS significantly associated with burnout after adjusting for covariates. Previous studies have shown the adverse effects of increased workloads among healthcare workers, manifesting burnout [52, 53]. In our study, we demonstrated that this association holds for AHPs in Singapore. In particular, the association of heavier workload among AHPs with a high burnout level is most apparent when the workload interferes with their "personal interests" and "work that must be done."

Hence, the identified associated factors of burnout levels highlight the need to address potential stressors at work. Concurrently, given that heavy self-reported workload and more extended work experience is associated with a high burnout level, workplace interventions are crucial. Based on this study, the association of heavier self-reported workload among AHPs with a high burnout level is most apparent when the workload becomes excessive or interferes with their interests. We propose that future studies look at interventions conducted at both personal and workplace levels [54].

Evidence-based strategies have shown the effectiveness of interventions that target personal coping skills such as mindfulness and stress management training [55, 56], and cognitive-behavioral interventions in reducing occupational stress levels [57].

Workplace strategies could be explored in future studies. Protected time, proper shift allocations, flexibility in working structure, and adequate workforce distribution could be highly beneficial [58, 59]. A case example will be the United Kingdom-commissioned review [60]. The review proposes a whole-system workplace intervention, from understanding local staff requirements, multi-level staff engagement, strong visible leadership, support for well-being at board level, and a focus on management capability to improve mental well-being and lower burnout.

Lastly, among participants who completed the PBPT questionnaire and experienced a high burnout level, 'negative evaluation of therapy' and 'time constraints' were identified as the most frequently cited barriers to seeking psychological help. Firstly, negative evaluation of therapy may be attributed to the high prevalence of negative attitudes towards mental illnesses in Asian societies such as Singapore [61, 62]. Participants may experience similar negative perceptions of therapy for mental health. Hence, interventions in improving the public perception towards mental health and therapy may reduce barriers to seeking help. Secondly, time constraints highlight that the daily responsibilities of AHPs may contribute to burnout and compete for time, hence a barrier in undergoing therapy. Daily responsibilities include formal duties to their patients and adjunct activities such as documentation, communication, following up on treatment, performing roll calls, or handing over. These auxiliary activities underestimate the time spent on the job [63]. Accounting for the adjunct activities and enforcing stricter regulations in total work hours may be essential to improve uptake of AHPs in seeking help for their burnout.

## Study limitations

There are a few limitations to this study. First, the response rate to the survey was only 29.1%. The low response rate may translate to a significant non-response bias for the study. Despite utilizing approaches to increase the response rate, such as through the engagement of respective departmental heads and email reminders, the survey response remained low. The low response rate may be due to hospital privacy protocols that limited the survey administration to emails and prevented physical surveys. Second, burnout is multi-factorial, and this study may not capture the full spectrum of variables. Factors that were not covered in this study include the increasing computerization of practice [64] and the participants' personality traits [65]. Third, this study's cross-sectional nature does not allow the authors to determine causal relationships between the risk factors and burnout. Further longitudinal studies will be needed. Fourth, other inventories such as the Copenhagen Burnout Inventory can be explored in future studies to offer new insights into burnout [66]. Fifth, as there are limited validation studies of MBI in Asian countries, MBI may have limited validity in characterizing burnout as a self-reported tool. Lastly, participant response could have been influenced by social desirability bias due to the highly stigmatized perception of burnout in the workplace.

## Conclusions

This study is the first to show a high burnout level and identify its associated factors among AHPs in Singapore. The self-reported burnout level among AHPs in this study was 67.4%. The identified risk factors included increased self-reported workload, lesser work experience, and younger age. Besides, respondents with a high burnout level reported the lack of motivation and time constraints as significant barriers to seeking psychological help for burnout. The findings revealed the significance and urgency of addressing burnout in these vulnerable target groups. There is also a potential need to implement individual and organizational interventions such as mindfulness and stress management training, cognitive-behavioral interventions, or workplace interventions that target organizational, cultural, social, and physical aspects of staff health. These interventions should be implemented with proper consideration of the barriers to reduce burnout risk effectively. Further longitudinal studies will help explore the causal relationship between the risk factors and burnout to characterize burnout's nature better.

## Supporting information

**S1 Data.**
(XLSX)

## Acknowledgments

The authors would like to thank the faculty members of Saw Swee Hock School of Public Health and the Yong Loo Lin School of Medicine, whose advice and ideas were integral to this study's success. The authors would like to thank the anonymous reviewers whose input and feedback significantly improved this manuscript. Lastly, the authors would like to thank the Community Health Project team members for their contributions to the study's conceptualization.

## Author Contributions

**Conceptualization:** Yao Hao Teo, Jordan Thet Ke Xu, Cowan Ho, Benjamin Kye Jyn Tan, Elisabeth Ker Hsuen Tan, Wei-An Goh, Elson Neo, Jonathan Yu Jing Chua, Sean Jun Yi Ng,

Julia Jie Yi Cheong, Jeff Yi-Fu Hwang, See Ming Lim, Thomas Soo, Judy Gek Khim Sng, Siyan Yi.

**Data curation:** Yao Hao Teo, Jui Min Leong.

**Formal analysis:** Yao Hao Teo, Jordan Thet Ke Xu, Jui Min Leong.

**Funding acquisition:** Jordan Thet Ke Xu, Jeff Yi-Fu Hwang, See Ming Lim, Thomas Soo, Judy Gek Khim Sng.

**Investigation:** Yao Hao Teo, Jordan Thet Ke Xu, Cowan Ho, Jui Min Leong, Benjamin Kye Jyn Tan, Elisabeth Ker Hsuen Tan, Wei-An Goh, Elson Neo, Jonathan Yu Jing Chua, Sean Jun Yi Ng, Julia Jie Yi Cheong, Jeff Yi-Fu Hwang, See Ming Lim, Thomas Soo, Judy Gek Khim Sng, Siyan Yi.

**Methodology:** Yao Hao Teo, Jordan Thet Ke Xu, Cowan Ho, Jui Min Leong, Benjamin Kye Jyn Tan, Elisabeth Ker Hsuen Tan, Wei-An Goh, Elson Neo, Jonathan Yu Jing Chua, Sean Jun Yi Ng, Julia Jie Yi Cheong, Jeff Yi-Fu Hwang, See Ming Lim, Thomas Soo, Judy Gek Khim Sng, Siyan Yi.

**Project administration:** Yao Hao Teo, Cowan Ho, Benjamin Kye Jyn Tan, Elisabeth Ker Hsuen Tan, Wei-An Goh, Elson Neo, Jonathan Yu Jing Chua, Sean Jun Yi Ng, Julia Jie Yi Cheong, Siyan Yi.

**Resources:** Jordan Thet Ke Xu, Cowan Ho, Jui Min Leong, Benjamin Kye Jyn Tan, Elisabeth Ker Hsuen Tan, Wei-An Goh, Elson Neo, Sean Jun Yi Ng, Julia Jie Yi Cheong, Jeff Yi-Fu Hwang, See Ming Lim, Thomas Soo, Judy Gek Khim Sng, Siyan Yi.

**Supervision:** Cowan Ho, Elisabeth Ker Hsuen Tan, Wei-An Goh, Elson Neo, Jonathan Yu Jing Chua, Sean Jun Yi Ng, Jeff Yi-Fu Hwang, See Ming Lim, Thomas Soo, Judy Gek Khim Sng, Siyan Yi.

**Validation:** Jui Min Leong, Benjamin Kye Jyn Tan, Wei-An Goh, Elson Neo, Jonathan Yu Jing Chua, Sean Jun Yi Ng, Jeff Yi-Fu Hwang, See Ming Lim, Thomas Soo, Judy Gek Khim Sng, Siyan Yi.

**Visualization:** Cowan Ho, Jeff Yi-Fu Hwang, See Ming Lim, Thomas Soo, Judy Gek Khim Sng, Siyan Yi.

**Writing – original draft:** Yao Hao Teo, Jordan Thet Ke Xu, Cowan Ho, Jui Min Leong, Benjamin Kye Jyn Tan, Elisabeth Ker Hsuen Tan, Wei-An Goh, Elson Neo, Jonathan Yu Jing Chua, Sean Jun Yi Ng, Julia Jie Yi Cheong.

**Writing – review & editing:** Yao Hao Teo, Jordan Thet Ke Xu, Cowan Ho, Jui Min Leong, Benjamin Kye Jyn Tan, Jeff Yi-Fu Hwang, See Ming Lim, Thomas Soo, Judy Gek Khim Sng, Siyan Yi.

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
