## [Decision Letter · Decision Letter 0]

18 Sep 2020

PONE-D-20-26669

Prevalence and associated factors of burnout in allied healthcare professionals in a tertiary hospital in Singapore

PLOS ONE

Dear Dr. Yi,

Thank you for submitting your manuscript to PLOS ONE. After careful consideration, we feel that it has merit but does not fully meet PLOS ONE’s publication criteria as it currently stands. Therefore, we invite you to submit a revised version of the manuscript that addresses the points raised during the review process.

We look forward to receiving your revised manuscript.

Kind regards,

Jenny Wilkinson, PhD

Academic Editor

PLOS ONE

Journal Requirements:

2.We note that you have indicated that data from this study are available upon request. PLOS only allows data to be available upon request if there are legal or ethical restrictions on sharing data publicly. For information on unacceptable data access restrictions, please see http://journals.plos.org/plosone/s/data-availability#loc-unacceptable-data-access-restrictions.

Additional Editor Comments (if provided):

Thank you for your submission, three reviewer reports have been received and I now invite you to provide a revision based on their comments. In particularly, work is needed on methodological aspects of the work and providing further explanation for readers in this area.

Reviewers' comments:

Reviewer's Responses to Questions

**Comments to the Author**

1. Is the manuscript technically sound, and do the data support the conclusions?

Reviewer #1: Yes

Reviewer #2: No

Reviewer #3: Partly

2. Has the statistical analysis been performed appropriately and rigorously? 

Reviewer #1: Yes

Reviewer #2: No

Reviewer #3: Yes

3. Have the authors made all data underlying the findings in their manuscript fully available?

Reviewer #1: Yes

Reviewer #2: Yes

Reviewer #3: No

4. Is the manuscript presented in an intelligible fashion and written in standard English?

Reviewer #1: Yes

Reviewer #2: Yes

Reviewer #3: No

5. Review Comments to the Author

Reviewer #1: The authors are congratulated for their work, which has been properly written and structured. To be able to publish it, it would be advisable that the authors could take into account the following minor comments:

The introduction is correct, but it can be expanded a little more. What is known about these job positions in other countries? What consequences have been observed in other countries with respect to burnout? What specific functions do they perform and why do they exist in some countries and not in others? What is their specific role in Singapore?

In reference 19, "In Singapore" is repeated twice. Study hypotheses are required.

Method: The type of mental illness must be described.

Discussion: What is the reason for a low response rate?

Reviewer #2: Thank you very much for allowing me to review this interesting and important study. I fully agree that there is too little in this area yet, in Singapore and beyond. This being said, I do have a number of concerns about your methods and interpretation. I have tried to order them in importance.

As you (partly) state in the limitations section, your study suffers from a low response rate. In addition, I am not convinced by the validity evidence you cite for all three scales. For these reasons, I advise you to be very careful with the word "prevalence". But let me go into more detail on the two points first, before elaborating why.

In regards to the former, do you have any information on the distribution of demographic characteristics in the target population? Does the sample reflect the target population in key characteristics, or were eg females more likely to respond to your email? This would be valuable information to judge on the representativeness of the sample despite the low response rate. In the limitations section, you state that "The low response rate may lead to a falsely lower burnout rate in our study." Kindly add an explanatory sentence. Could it not also be the other way around, ie people with burnout symptoms being more prone to answering a survey which speaks to their own experiences?

In regards to the latter, are there no validation studies in Singapore or at least in a sligthly "closer" linguistic and cultural context, especially for the MBI? As for the alpha's that you report from the referenced studies, I'm not sure there is much value in that, especially since you present your own alpha's in the results section (although I'm not quite sure why also for PA, which you don't use), which are much more meaningful.

Assuming there are not well-fitting and robust validation studies, it would be good to explicitly address the issue in the limitation section. Plenty of literature has shown that many self-reported screening tools developed in Europe or North America have only limited validity in other parts of the world due to differences in experiences, differences in answer behavior, etc.

For the above two reasons, I advise you to be very careful with the word "prevalence". You use a non-validated self-reported screening tool in a population with low response rate, so if anything, your results will be in the ball park of the "actual" burnout situation, but likely quite a bit off (as it would be evaluated by a clinical psychologist). You could for instance write "68% reported symptoms indicative of burnout" or "68% had high scores" instead. Just to be very clear - I do not mean to invalidate your study in any way, quite the opposite, but find it important to make the limitations very clear to readers without background in epidemiology and diagnostics.

Given that you use only two dimensions to define burnout, I would further be careful with presenting an overall prevalence. This will be confusing to readers trying to compare across contexts, as the "norm" is use of all three subscales for an overall burnout prevalence. I speak from the position of somebody who has recently tried to perform a meta-analysis with MBI studies, and ended up extremely frustrated as comparability was very difficult due to this element and others. I therefore think it would be better and clearer to report on the two dimensions separately.

It would further be good if you added a sentence or two to the description of each measure in the methods section allowing the reader to understand how you arrived at the numeric values, and where to situate your cut-off scores on the maximum range of scores. How many items per dimension, which response scale did you use, how was it scored, how did you combine the individual items, and what was the final theoretical range per domain? Again, this was the main reason why I couldn't include most studies in the meta-analysis. This is particularly important for AWS, as it's impossible to make much sense of Figures 2 and 3 without this background information, but also to be able to put into perspective (and be able to compare across settings) the burnout cutoff scores. For burnout, since in Figure 1 you report low, moderate, high, it would also be good to give thresholds for low-moderate, not just for moderate-high.

In terms of results, I'm not sure I find what you did in Table 3 meaningful, and I don't agree with the conclusions you draw on its basis. Since the measure is about psychological help related to burnout, why do you compare people with burnout with people without, rather than to examine the frequency of cited factors in the group of respondents with burnout? For instance, you conclude that "lack of motivation" is associated with burnout, but in the sense that people without burnout have more frequent lack of motivation to seek help for burnout - which is not surprising since they also don't have need! The much more interesting information to me is that in the burned out group, 1/4 experience lack of motivation to seek help (if I read the proportion right). Consider reorganizing the table and aligning the interpretation. In the discussion, you write that "‘lack of

motivation’ and ‘time constraints’ were identified as significant barriers to seeking psychological help.". However, if I read the table right, the most frequently cited barriers in the burned out group are actually "negative evaluation of therapy", "time constraints", and "stigma".

And finally, a few more minor comments:

In terms of organization of results, it's a bit unconventional to mix sample description and substantial results - I suggest you separate this (effectively removing the sentence on burnout prevalence by cadre).

Table 1 seems is missing the non-Singaporeans, females, non-Chinese ethnic, caregiver-non, not being frontline staff, not being employed full time, high level of physical activity, no history of mental illness, and no help seeking in last year. The way I read the proportions, proportions of participants with and without burnout cannot easily be calculated for them from the respective other groups (one would for instance have to subtract the burned out males from all burned out participants and same for the non-burned-out to then know how many females and how they are distributed among burned out and not burned out).

In the methods, you say that you didn't use PA, but you present the data in Figure 1. Kindly align or clarify. For PA, did you reverse scores, ie Figure 1 high refers to low personal accomplishment (indicative of burnout)? Or is high high personal accomplishment (ie no burnout)?

Discussion - differences in DP prevalences US-Singapore: You attribute this to the setting, but it might well be to differences in responding to items. Since there seems to be no robust validation/equivalence study, I would at least point out that it might also just be a methodological artefact, or explicitly state the assumption of cultural equivalence.

"Compared to younger employees, older workers have been better at handling occupational stress and are less prone to burnout [43]." Consider revising this strong statement (eg "A study in Iran, for instance, has found (or just speculated?) that older workers might be better at handling occupational stress and thereby be less prone to burnout). If this is a "proven fact" (by many robust studies), then kindly cite accordingly.

Similarly, kindly be careful with statements implying causality. For instance, on the association workload - burnout. Since all is self-report to my understanding (and not even self-reported facts such as number of work hours), could it not also be that burned out people perceive their workload to be higher?

I don't fully understand how you infer "In particular, the association of heavier workload among burnout AHPs is most apparent when the workload interferes with their “personal interests” and “work that must be done.”" Kindly explain.

Reviewer #3: The paper focuses on burnout and refers in particular to healthcare professionals. Burnout is one of the most studied variables in healthcare and there is established literature on it. Since the World Health Organization has included burnout in the revision of the ICD-11 as an occupational phenomenon, this research topic is still current and interesting.

Title: I suggest to avoid the term “in allied”

Abstract: Results: the authors say “Burnout was positively associated with a longer work experience of 3 to 5 years” Do you think 3-5 years is a longer work experience?. Also, authors state that “and more than five years (AOR 4.24; 95% CI 1.16-16.79)”, unclear 5 years of experience in the profession or position?

In abstract authors say that, “there is a lack of studies on burnout among allied health professionals in Singapore”, but in introduction you state “For example, high rates of burnout and its associations among physicians and nurses have been reported in Singapore. Extensive research involves the barriers to seeking help for doctors, such as fear of stigma, lack of available time, and lack of convenient access” and latter “However, there are currently no studies examining the prevalence and associations of burnout among AHPs in Singapore, and also in discussion “This study is the first to investigate the prevalence of burnout and its related factors among AHPs in Singapore” Please reconcile, these sentences are contradictory”.

Why the authors use the term “Allied health professionals” and not only healthcare professionals ? There is any difference

This sentence is not clear: “In Singapore, In Singapore, the Allied Health Professions Council (AHPC) further classified allied health occupations”

Study design and sampling: “among AHPs”. Clarify what kind of professionals?

“between October 2019 to December 2019”. Did the survey coincide with COVID-19 pandemic?

I am not clear, what is the difference between “Staff members in the tertiary hospital” and “AHP”, also authors state “definition established by the AHPC in Singapore”.

I’m not clear, you say “Among the three subscales, PA was excluded from this study because its association with burnout has been more variable and complex, similar to previous studies”, the MBI includes all 3 dimensions. If the authors do not believe that this instrument was valid, why did they not use another measurement instrument such as the Copenhagen Burnout Inventory?

Questionnaire design and measurement: please establish the cut-off score of low, medium and high levels in the three dimensions, and not only the cut-off for high EE and D, and low PA.

In table 1. Columns “With burnout and Without burnout” are not clear, what criteria did the authors follow to consider burnout, if they did not take PA into account?

Also authors say: “The Cronbach’s alpha coefficients for EE, DP, and PA in MBI-HSS were 0.93, 0.81, and 0.85, respectively, suggesting that the overall measurement was reliable”, Is this data your this study? I understand that you collected PA data but you did not analyse it?

Discussion: “workplace interventions are crucial”. Please expand what kind of interventions could be carried out

Editing by a written English expert would improve word choice and overall flow.

6. PLOS authors have the option to publish the peer review history of their article (what does this mean?). If published, this will include your full peer review and any attached files.

Reviewer #1: No

Reviewer #2: No

Reviewer #3: No

---

## [Author Response · Author response to Decision Letter 0]

3 Nov 2020

Responses to reviewers (R1)

PONE-D-20-26669

Prevalence and associated factors of burnout in allied healthcare professionals in a tertiary hospital in Singapore

PLOS ONE

Editorial Comments:

RESPONSE: Thank you for the reminder. We have ensured that the manuscript meets PLOS ONE’s style and requirements, including file naming.

RESPONSE: 

We have uploaded the anonymized data set as a Supporting Information file.

Academic Editor: Dr. Jenny Wilkinson, PhD

Thank you for your submission, three reviewer reports have been received and I now invite you to provide a revision based on their comments. In particularly, work is needed on methodological aspects of the work and providing further explanation for readers in this area.

RESPONSE: 

We would like to take this opportunity to thank the Academic Editor for your kind consideration of our manuscript and your guidance for improving it. We have addressed all reviewers’ comments very carefully and provided a point-by-point response to each comment. We hope you will agree that the manuscript has been significantly improved and could be published in PLOS ONE.

Reviewer #1

The authors are congratulated for their work, which has been properly written and structured. To be able to publish it, it would be advisable that the authors could take into account the following minor comments:

RESPONSE: 

We thank the reviewer for your constructive feedback. We have made the changes according to the reviewer’s comments where possible and provided a point-by-point reply below, which we hope will address the reviewer’s feedback.

1. The introduction is correct, but it can be expanded a little more. What is known about these job positions in other countries? What consequences have been observed in other countries with respect to burnout? What specific functions do they perform and why do they exist in some countries and not in others? What is their specific role in Singapore?

RESPONSE: 

Thank you for your comment. Your suggestions have been incorporated in Page 3 Paragraph 4 under Introduction. We have also expanded on the various points as follows.

For our definition of Allied Health Professionals, we have referred to the definition provided by the US Association of Schools Advancing Health Professions. This definition refers to Allied Health Professionals as “individuals, distinct from medicine and nursing, who work collaboratively to deliver routine and essential healthcare services. They include but are not limited to occupational therapists, physiotherapists, pharmacists, medical social workers, and radiographers.” We hope this definition helps to improve the clarity of the Introduction.

Thank you for pointing out the importance of describing the specific role Allied Health Professionals play in Singapore compared to other countries. In Singapore, the Allied Health Professions Council regulates the "professional standards for practice, conduct & ethics of registered allied health professionals in Singapore.” Their definition of an Allied Health Professional is similar to that stated above and includes similar groups of healthcare professionals, such as occupational therapists and physiotherapists.

Beyond Singapore, Allied Health Professionals play similar roles in the United Kingdom and the United States. They improve hospital efficiency and access to healthcare and promote better health-related quality of life. This is again similar to the definition provided by the US Association of Schools Advancing Health Professions. 

Thank you for pointing out the importance of stating the consequences of burnout in other countries as well. We have incorporated it on Page 3, Paragraph 2. Previous studies have demonstrated a wide variety of consequences that we have divided into individual and institutional categories. These studies showed that burnout had been observed to be associated with increased absenteeism, medical errors, sickness, injury and even accidents on an individual level. On an institutional level, burnout was linked to a higher turnover of healthcare workers, decreased workforce efficiency, and higher economic burden on the healthcare system. These consequences were reported in burnout studies done on populations in the United States, Poland, and Switzerland. 

2. In reference 19, "In Singapore" is repeated twice.

RESPONSE: 

Thank you for pointing it out. We sincerely apologize for the typo. We have removed the repeated “In Singapore.”

3. Study hypotheses are required.

RESPONSE: 

Thank you for your feedback. We have included the hypothesis in the manuscript on Page 4, Paragraph 2. Our study was conducted with the hypothesis that there will be a high level of burnout among Allied Health Professionals in Singapore. 

As there have been no studies of burnout among Allied Health Professionals in Singapore, we based this hypothesis on the existing burnout studies done on the physician and nursing populations in Singapore. These studies found a significant self-reported burnout prevalence at 30-40% of the study population. We thus hypothesized that our study results would be similar due to shared workplace challenges and occupation settings in healthcare.

The study by Dhaliwal et al. on burnout among nurses in Singapore found “age” and “job grade” were significantly associated with burnout defined by MBI. Similarly, W. Y Tay reported “increased number of years working as a nurse [...] to be significantly associated with burnout.” It was in reference to these studies we based our hypothesis. 

4. Method: The type of mental illness must be described.

RESPONSE: 

Thank you for your comment. We agree that there is a need for a specific description of mental illness. We chose to phrase mental illness as a broader classification to accept any type of mental illness. We have included in the footnote of Table 1 on Page 9, defining mental illness as “a behavioral or psychological syndrome or pattern that occurs in an individual which causes clinically significant distress. It warrants diagnosis and management by a medical professional”. Nonetheless, we understand that this might pose ambiguity to respondents and hence our survey participants were all provided contact details of the Principal Investigators should they have any queries.

5. Discussion: What is the reason for a low response rate?

RESPONSE: 

Thank you for your comment. Our study team initiated various strategies to increase the response rate. Firstly, we approached the healthcare institution’s administrative coordinators and worked with them to disseminate the survey to their staff. The survey was emailed to them, and they were actively encouraged and reminded to take part in it. Secondly, we coordinated with public health professionals from the Saw Swee Hock School of Public Health and sought their prior experience in improving recruitment. Finally, we worked with the leadership of allied health departments from the National University Health System to optimize the reach of our emails.

We understand that physically administering the survey might have generated a higher response rate. However, hospital privacy guidelines prevented access to workplaces and the conduction of physical surveys. 

We acknowledge that this low response rate has potentially introduced bias into our study. We have included this point under Study Limitations on pages 16-17, stating that the low response rate translated to a ‘significant non-response bias for the study.’ Nevertheless, we think these findings may still be relevant to policymakers because of the dearth of studies on burnout in Allied Health Professionals. It provides a starting point for future, more comprehensive studies to characterize the true extent of the burnout problem and explore potential interventional strategies for burnout.

Reviewer #2

Thank you very much for allowing me to review this interesting and important study. I fully agree that there is too little in this area yet, in Singapore and beyond. This being said, I do have a number of concerns about your methods and interpretation. I have tried to order them in importance.

As you (partly) state in the limitations section, your study suffers from a low response rate. In addition, I am not convinced by the validity evidence you cite for all three scales. For these reasons, I advise you to be very careful with the word "prevalence". But let me go into more detail on the two points first, before elaborating why.

RESPONSE: 

We thank the reviewer for your insights and constructive feedback. We have provided a point-by-point reply below and incorporated the reviewer’s suggestions where possible, which we hope will address the reviewer’s concerns.

1. In regards to the former, do you have any information on the distribution of demographic characteristics in the target population? Does the sample reflect the target population in key characteristics, or were eg females more likely to respond to your email? This would be valuable information to judge on the representativeness of the sample despite the low response rate. In the limitations section, you state that "The low response rate may lead to a falsely lower burnout rate in our study." Kindly add an explanatory sentence. Could it not also be the other way around, ie people with burnout symptoms being more prone to answering a survey which speaks to their own experiences?

RESPONSE: 

Thank you for pointing this out. We agree with the comments. Therefore, we have included statements comparing the demographics of respondents with non-respondents. (Page 7, paragraph 1) In summary, compared to those who did not participate, those who participated were more likely to be female, non-Singaporeans/non-SPR, 21 to 30 years old, and had more than 3 years of working experience. 

We also agree that our inference in the limitations section is an unsupported statement. We agree that the low response rate may lead to either a falsely lower or higher burnout rate. Hence, we have removed the statement.

2. In regards to the latter, are there no validation studies in Singapore or at least in a slightly "closer" linguistic and cultural context, especially for the MBI? 

As for the alpha's that you report from the referenced studies, I'm not sure there is much value in that, especially since you present your own alpha's in the results section (although I'm not quite sure why also for PA, which you don't use), which are much more meaningful.

Assuming there are not well-fitting and robust validation studies, it would be good to explicitly address the issue in the limitation section. Plenty of literature has shown that many self-reported screening tools developed in Europe or North America have only limited validity in other parts of the world due to differences in experiences, differences in answer behavior, etc.

RESPONSE: 

Thank you for your comment. There have been limited validation studies in Asian societies in our literature review. To address this issue, we highlighted how the MBI had been used in studies conducted in China, Japan, and Singapore. These include Wang Z et al., Nishimura K et al., and See KC, respectively.

Thank you for pointing out the value in reporting the Cronbach alpha’s of the referenced articles. We agree with you on only presenting the more meaningful values from our studies. Hence, we have removed the Cronbach alpha’s of the referenced studies.

We agree with your comment on explicitly addressing a lack of validation studies in the limitation section. On Page 17, Paragraph 2, we have included this limitation, stating, ‘as there are limited validation studies of MBI in Asian countries, MBI may have limited validity in characterizing burnout as a self-reported tool.’

3. For the above two reasons, I advise you to be very careful with the word "prevalence". You use a non-validated self-reported screening tool in a population with low response rate, so if anything, your results will be in the ball park of the "actual" burnout situation, but likely quite a bit off (as it would be evaluated by a clinical psychologist). You could for instance write "68% reported symptoms indicative of burnout" or "68% had high scores" instead. Just to be very clear - I do not mean to invalidate your study in any way, quite the opposite, but find it important to make the limitations very clear to readers without background in epidemiology and diagnostics.

Given that you use only two dimensions to define burnout, I would further be careful with presenting an overall prevalence. This will be confusing to readers trying to compare across contexts, as the "norm" is use of all three subscales for an overall burnout prevalence. I speak from the position of somebody who has recently tried to perform a meta-analysis with MBI studies, and ended up extremely frustrated as comparability was very difficult due to this element and others. I therefore think it would be better and clearer to report on the two dimensions separately.

RESPONSE: 

Thank you for your advice on using the word ‘prevalence.’ We agree with your comment and have incorporated your suggestion throughout the manuscript. The use of ‘prevalence’ has been changed to ‘self-reported levels’ in the title, abstract, and throughout the manuscript.

Thank you for your suggestion. We have reported the levels of burnout according to the 2 dimensions separately (Page 9, Paragraph 1), stating ‘A majority of the respondents reported high level of burnout on EE (n=203, 61.9%), less than half reported a high level on DP (n=139, 42.4%), and more than one-third had both high EE and DP (n=122, 37.1%).’ 

Thank you for pointing this out. We have decided to report on the overall self-reported burnout levels according to the pre-specified cut-offs of EE>27 and/or DP>10 based on the following reasons. While there is no universal cut-off score to define burnout, in a recent systematic review of burnout in healthcare professionals, burnout was defined using the cut-offs of only EE>27 and/or DP>10, with PA excluded, in the majority of participants. Furthermore, it has been postulated by Maslach that PA’s association has been more variable and complex, and it may be a function of EE and DP instead. Hence, we defined burnout according to the cut-offs of EE>27 and/or DP>10. We have included this explanation in the methodology to minimize confusion for the readers (Page 5, Paragraph 2). 

4. It would further be good if you added a sentence or two to the description of each measure in the methods section allowing the reader to understand how you arrived at the numeric values, and where to situate your cut-off scores on the maximum range of scores. How many items per dimension, which response scale did you use, how was it scored, how did you combine the individual items, and what was the final theoretical range per domain? Again, this was the main reason why I couldn't include most studies in the meta-analysis. This is particularly important for AWS, as it's impossible to make much sense of Figures 2 and 3 without this background information, but also to be able to put into perspective (and be able to compare across settings) the burnout cutoff scores. For burnout, since in Figure 1 you report low, moderate, high, it would also be good to give thresholds for low-moderate, not just for moderate-high.

RESPONSE: 

Thank you for the suggestion. We agree and have added descriptions for MBI and AWS, including number of items, dimensions, and the relevant information on scoring and cut-offs used under the “Questionnaire design and measurement” pages 4-6. For convenient reference, the cut offs for low, moderate, high used in Figure 1 previously are as follows:

EE High: >27, Moderate: 19-26, Low 0-18

DP High: >10, Moderate: 6-9, Low: 0-5

PA High: 0-33, Moderate: 34-39, Low: >40

However, we have removed the individual low, moderate, high cutoffs, and the original figure 1 from the paper to improve clarity to readers, as the cutoffs were not meaningful for discussion in the subsequent analyses.

5. In terms of results, I'm not sure I find what you did in Table 3 meaningful, and I don't agree with the conclusions you draw on its basis. Since the measure is about psychological help related to burnout, why do you compare people with burnout with people without, rather than to examine the frequency of cited factors in the group of respondents with burnout? For instance, you conclude that "lack of motivation" is associated with burnout, but in the sense that people without burnout have more frequent lack of motivation to seek help for burnout - which is not surprising since they also don't have need! The much more interesting information to me is that in the burned out group, 1/4 experience lack of motivation to seek help (if I read the proportion right). Consider reorganizing the table and aligning the interpretation. In the discussion, you write that "‘lack of

motivation’ and ‘time constraints’ were identified as significant barriers to seeking psychological help." However, if I read the table right, the most frequently cited barriers in the burned out group are actually "negative evaluation of therapy", "time constraints", and "stigma".

RESPONSE: 

Thank you for pointing this out. We agree with this comment. Therefore, we have reorganized the table and only analyzed (Page 14, Paragraph 1) and discussed (Page 16, Paragraph 4) the frequently cited barriers to seeking psychological help among participants who experienced high levels of burnout. For convenient reference, among burnout participants, the most frequently cited barriers to seeking psychological help were ‘negative evaluation of therapy’ (60%), and ‘time constraints’ (50%). 

6. In terms of organization of results, it's a bit unconventional to mix sample description and substantial results - I suggest you separate this (effectively removing the sentence on burnout prevalence by cadre).

RESPONSE: 

Thank you for your suggestion. We have separated it into Table 1 (sample description) and Table 2 (substantial results).

7. Table 1 seems is missing the non-Singaporeans, females, non-Chinese ethnic, caregiver-non, not being frontline staff, not being employed full time, high level of physical activity, no history of mental illness, and no help seeking in last year. The way I read the proportions, proportions of participants with and without burnout cannot easily be calculated for them from the respective other groups (one would for instance have to subtract the burned out males from all burned out participants and same for the non-burned-out to then know how many females and how they are distributed among burned out and not burned out).

RESPONSE: 

Thank you for your comment. We agree that it is difficult to be calculated and is confusing for readers. Therefore, we have added the missing data into Tables 1 and 2, and we hope this improves clarity for readers.

8. In the methods, you say that you didn't use PA, but you present the data in Figure 1. Kindly align or clarify. For PA, did you reverse scores, ie Figure 1 high refers to low personal accomplishment (indicative of burnout)? Or is high personal accomplishment (ie no burnout)?

RESPONSE: 

Thank you for your comment. By convention, low PA scores in the MBI correspond to high burnout. The cut-offs are as follows:

EE High burnout: >27, Moderate: 19-26, Low 0-18

DP High burnout: >10, Moderate: 6-9, Low: 0-5

PA High burnout: 0-33, Moderate: 34-39, Low: >40

We have removed the cut-offs, PA, and the original figure 1 from the paper to improve clarity to readers, as they are not meaningful for discussion in the subsequent analyses. The rationale for why PA was not used is explained in the methodology (Page 5, paragraph 2), and question 3 of the reviewer’s comments. 

9. Discussion - differences in DP prevalence US-Singapore: You attribute this to the setting, but it might well be to differences in responding to items. Since there seems to be no robust validation/equivalence study, I would at least point out that it might also just be a methodological artefact, or explicitly state the assumption of cultural equivalence.

RESPONSE: 

Thank you for your comment. We agree with your suggestion. We acknowledge that there are inherent cultural differences between the two populations. This is supported by previous studies. Dolnicar, S. et al. stated that a difference in the cultural background is a “significant potential source of misinterpretation in cross-cultural studies.” We agree that this would affect the response to the surveys and the study’s results, especially with the differences in the concept of depersonalization and work by our study participants. Hence, we have assumed that there is cultural equivalence to be able to compare the results. 

10. "Compared to younger employees, older workers have been better at handling occupational stress and are less prone to burnout [43]." Consider revising this strong statement (eg "A study in Iran, for instance, has found (or just speculated?) that older workers might be better at handling occupational stress and thereby be less prone to burnout). If this is a "proven fact" (by many robust studies), then kindly cite accordingly.

RESPONSE: 

Thank you for the comment. We agree and have revised the statement accordingly for greater clarity. The revised sentence now reads, “The lower burnout level in older participants may be explained by their better coping or occupational handing of stress [51,52].” We have cited the following two studies: Scheibe et al. reported an “older-age advantage for recovery from work-demands,” and Hsu HC, who reported on the increased “resilience of older workers.”

11. Similarly, kindly be careful with statements implying causality. For instance, on the association workload - burnout. Since all is self-report to my understanding (and not even self-reported facts such as number of work hours), could it not also be that burned out people perceive their workload to be higher?

RESPONSE: 

Thank you for pointing it out. We have accordingly revised the word ‘contributed’ to ‘associated with.’ We have also changed ‘workload’ to ‘self-reported workload’ to improve clarity in the abstract, discussion, and conclusion.

12. I don't fully understand how you infer "In particular, the association of heavier workload among burnout AHPs is most apparent when the workload interferes with their “personal interests” and “work that must be done.”" Kindly explain.

RESPONSE: 

Thank you for your comment. ‘Personal interests’ and ‘work that must be done’ refer to questions 3 and 1 of the AWS questionnaire, respectively. Both questions are part of the workload domain and have the highest absolute mean difference between burnout and non-burnout groups (Figure 3) among all questions. Hence, this association of heavier workload was noted to be most apparent in the context of these individual statements (part of the workload domain). We have added this explanation in the results to improve clarity (Page 12, Paragraph 2).

Reviewer #3

The paper focuses on burnout and refers in particular to healthcare professionals. Burnout is one of the most studied variables in healthcare and there is established literature on it. Since the World Health Organization has included burnout in the revision of the ICD-11 as an occupational phenomenon, this research topic is still current and interesting.

RESPONSE: 

We thank the reviewer for your constructive feedback. We have incorporated the reviewer’s suggestions where possible and provided a point-by-point reply below, which we hope have addressed the reviewer’s concerns.

1. Title: I suggest to avoid the term “in allied”

RESPONSE: 

Thank you for the suggestion. After careful consideration, we have decided to use the term “Allied Health Professionals” throughout our paper, to distinguish from “Health Professionals,” which would include doctors and nurses. For our definition of Allied Health Professionals, we have referred to the definition provided by the US Association of Schools Advancing Health Professions, which define Allied Health Professionals as “individuals, distinct from medicine and nursing, who work collaboratively to deliver routine and essential healthcare services. They include but are not limited to occupational therapists, physiotherapists, pharmacists, medical social workers, and radiographers.” 

However, in consideration of your suggestion, we have made corrections to make the definition of “Allied Health Professionals” clearer to readers, both through corrections in our Abstract (Page 2, Paragraph 1) and Introduction (Page 3, Paragraph 4). We thank the reviewer for this opportunity to improve the clarity of our manuscript.

2. Abstract: Results: the authors say “Burnout was positively associated with a longer work experience of 3 to 5 years” Do you think 3-5 years is a longer work experience? Also, authors state that “and more than five years (AOR 4.24; 95% CI 1.16-16.79)”, unclear 5 years of experience in the profession or position?

RESPONSE: 

Thank you for the suggestion. We have accepted the edit and the sentence now reads, "Burnout was positively associated with a work experience of 3 to 5 years and more than 5 years, as compared to that of 1 to 2 years.” This can be found in the 3rd paragraph of the Abstract (Page 2 Paragraph 3)

3. In abstract authors say that, “there is a lack of studies on burnout among allied health professionals in Singapore”, but in introduction you state “For example, high rates of burnout and its associations among physicians and nurses have been reported in Singapore. Extensive research involves the barriers to seeking help for doctors, such as fear of stigma, lack of available time, and lack of convenient access” and latter “However, there are currently no studies examining the prevalence and associations of burnout among AHPs in Singapore, and also in discussion “This study is the first to investigate the prevalence of burnout and its related factors among AHPs in Singapore” Please reconcile, these sentences are contradictory”.

RESPONSE:

Thank you for the suggestion. We have accepted the suggestion and have made clearer the definition of “Allied Health Professionals” right from the start. The sentence now reads “Allied Health Professionals (AHPs) are defined as individuals, excluding physicians and nurses, who work collaboratively to deliver routine and essential healthcare services.”

This sentence can be found in the 1st paragraph of the Abstract (Page 2 Paragraph 1) and the 4th paragraph of the Introduction (Page 3 Paragraph 4).

4. Why the authors use the term “Allied health professionals” and not only healthcare professionals? There is any difference.

RESPONSE:

Thank you for your suggestion. We have clarified our definition of Allied Health Professionals, similar to the comment above. 

5. This sentence is not clear: “In Singapore, In Singapore, the Allied Health Professions Council (AHPC) further classified allied health occupations”

RESPONSE:

Thank you for your suggestion. We agree and have made the relevant changes. We have defined AHPs in the earlier part of that paragraph (Page 3 Paragraph 4). Hence, we have rephrased it to “In Singapore, the Allied Health Professions Council (AHPC) defines and classifies allied health occupations similar to that of other countries” to improve clarity that our definition of AHPs is similar to international definitions. We would also like to sincerely apologize for the typo that was present in the original sentence. 

6. Study design and sampling: “among AHPs”. Clarify what kind of professionals?

RESPONSE:

Thank you for your suggestion. We have clarified our definition of Allied Health Professionals. As we have earlier defined AHPs in the introduction, we similar included AHPs in the study design as “healthcare professionals, excluding physicians and nurses, who work collaboratively to deliver routine and essential healthcare services” on Page 4, Paragraph 4.

7. “between October 2019 to December 2019”. Did the survey coincide with COVID-19 pandemic?

RESPONSE: 

To clarify, the survey did not coincide with the COVID-19 pandemic. 

8. I am not clear, what is the difference between “Staff members in the tertiary hospital” and “AHP”, also authors state “definition established by the AHPC in Singapore”.

RESPONSE:

Thank you for your suggestion. We agree and have made the relevant changes. We have removed the lines “Definition established by the AHPC in Singapore,” as we have earlier fully defined AHPs in the 4th paragraph of the Introduction (Page 3, Paragraph 4). As such, the inclusion criteria on page 4 now reads, “We included AHPs according to the definition stated in the introduction - all healthcare professionals, excluding physicians and nurses, who work collaboratively to deliver routine and essential healthcare services. AHPs in a tertiary hospital of all seniority levels were included in this study,” with no reference to AHPC. 

9. I’m not clear, you say “Among the three subscales, PA was excluded from this study because its association with burnout has been more variable and complex, similar to previous studies”, the MBI includes all 3 dimensions. If the authors do not believe that this instrument was valid, why did they not use another measurement instrument such as the Copenhagen Burnout Inventory?

RESPONSE:

Thank you for the opportunity to clarify. The MBI originally came with all 3 subscales. While there is no universal cut-off score to define burnout, in a recent systematic review of burnout in healthcare professionals, burnout was defined using the cut-offs of only EE>27 and/or DP>10, with PA excluded, in the majority of participants. Furthermore, it has been postulated by Maslach that PA’s association has been more variable and complex, and it may be a function of EE and DP instead. Hence, we defined burnout according to the cut-offs of EE>27 and/or DP>10. We have included this explanation in the methodology to minimize confusion for the readers (Page 5, Paragraph 2). 

We agree that other measurement instruments exist. Our analysis suggests that the overall measurement using MBI was reliable. The Cronbach’s alpha coefficients for EE, DP, and PA in MBI-HSS were 0.93, 0.81, and 0.85, respectively (Page 7, paragraph 2). Also, it is the most widely used measurement instrument. Nevertheless, we thank you for your suggestion and believe that other inventories like the Copenhagen Burnout Inventory can be explored in future studies to offer new insights into burnout. We have included this in the 2nd last line of the “Study Limitations” section on Page 17, Paragraph 2.

10. Questionnaire design and measurement: please establish the cut-off score of low, medium and high levels in the three dimensions, and not only the cut-off for high EE and D, and low PA.

RESPONSE:

Thank you for your comment. The cut offs are as follows:

EE High: >27, Moderate: 19-26, Low 0-18

DP High: >10, Moderate: 6-9, Low: 0-5

PA High: 0-33, Moderate: 34-39, Low: >40

We have removed this and the original figure 1 from the paper to improve clarity to readers, as the cutoffs were not meaningful for discussion in the subsequent analyses.

11. In table 1. Columns “With burnout and Without burnout” are not clear, what criteria did the authors follow to consider burnout, if they did not take PA into account? Also authors say: “The Cronbach’s alpha coefficients for EE, DP, and PA in MBI-HSS were 0.93, 0.81, and 0.85, respectively, suggesting that the overall measurement was reliable”, Is this data your this study? I understand that you collected PA data but you did not analyze it?

RESPONSE: 

Thank you for pointing this out. We have decided to report on the overall self-reported burnout levels according to the pre-specified cut-offs of EE>27 and/or DP>10. Our rationale for using this cut-off is explained in question 8 of reviewer 3. For convenient reference: While there is no universal cut-off score to define burnout, in a recent systematic review of burnout in healthcare professionals, burnout was defined using the cut-offs of only EE>27 and/or DP>10, with PA excluded, in the majority of participants. Furthermore, it has been postulated by Maslach that PA’s association has been more variable and complex, and it may be a function of EE and DP instead. Hence, we defined burnout according to the cut-offs of EE>27 and/or DP>10. We have included this explanation in the methodology to minimize confusion for the readers on Page 5, Paragraph 2. 

12. Discussion: “workplace interventions are crucial”. Please expand what kind of interventions could be carried out.

RESPONSE: 

Thank you for the comment. We have expanded on interventions in our manuscript as per your suggestion. We have categorized our proposed interventions into personal and workplace suggestions and have expanded on each point within the manuscript. 

Given that heavy self-reported workload and more extended work experience are associated with burnout in our study, we expanded on the kind of workplace interventions could be explored in future studies. Referencing previous studies, possible workplace strategies include “Protected time, proper shift allocations, flexibility in working structure, and adequate manpower distribution”.

Thank you for your suggestion. We believe the revision per your comment has made our proposed interventions clearer and more actionable.

---

## [Editor Report · Decision Letter 1]

30 Nov 2020

PONE-D-20-26669R1

Factors associated with self-reported burnout level in allied healthcare professionals in a tertiary hospital in Singapore

PLOS ONE

Dear Dr. Yi,

Thank you for submitting your manuscript to PLOS ONE. After careful consideration, we feel that it has merit but does not fully meet PLOS ONE’s publication criteria as it currently stands. Therefore, we invite you to submit a revised version of the manuscript that addresses the points raised during the review process.

We look forward to receiving your revised manuscript.

Kind regards,

Jenny Wilkinson, PhD

Academic Editor

PLOS ONE

Additional Editor Comments (if provided):

Thank you for your responses to reviewers and accompanying manuscript changes. These has addressed the comments with one minor revision now needed. In the Abstract Results new abbreviations have been added which have not been previously explained (i.e. EE and DP)

---

## [Author Response · Author response to Decision Letter 1]

1 Dec 2020

Editor Comments:

Thank you for your responses to reviewers and accompanying manuscript changes. These has addressed the comments with one minor revision now needed. In the Abstract Results new abbreviations have been added which have not been previously explained (i.e. EE and DP).

RESPONSE: We apologized for the oversight. We have spelled out the abbreviations per the editor's advice. The sentence now read:

"The self-reported burnout level (emotional exhaustion>27 and/or depersonalization>10) was 67.4%."

Please see lines 42-43.

We have also proofread the revised manuscript with a few minor changes made. Please see the 'Revised manuscript with track changes_R2.'

---

## [Editor Report · Decision Letter 2]

8 Dec 2020

Factors associated with self-reported burnout level in allied healthcare professionals in a tertiary hospital in Singapore

PONE-D-20-26669R2

Dear Dr. Yi,

We’re pleased to inform you that your manuscript has been judged scientifically suitable for publication and will be formally accepted for publication once it meets all outstanding technical requirements.

Kind regards,

Jenny Wilkinson, PhD

Academic Editor

PLOS ONE
---

## [Editor Report · Acceptance letter]

14 Dec 2020

PONE-D-20-26669R2 

Factors associated with self-reported burnout level in allied healthcare professionals in a tertiary hospital in Singapore 

Dear Dr. Yi:

I'm pleased to inform you that your manuscript has been deemed suitable for publication in PLOS ONE. Congratulations! Your manuscript is now with our production department. 

Kind regards, 

on behalf of

Dr Jenny Wilkinson 

Academic Editor

PLOS ONE